# Damage Evolution Characteristics of Back-Filling Concrete in Gob-Side Entry Retaining Subjected to Cyclical Loading

**DOI:** 10.3390/ma15165772

**Published:** 2022-08-21

**Authors:** Xicai Gao, Shuai Liu, Cheng Zhao, Jianhui Yin, Kai Fan

**Affiliations:** 1State Key Laboratory of Coal Resources in Western, Xi’an University of Science and Technology, Xi’an 710054, China; 2Key Laboratory of Western Mine Exploitation and Hazard Prevention, Ministry of Education, Xi’an University of Science and Technology, Xi’an 710054, China; 3Shaanxi Coal and Chemical Technology Institute Co., Ltd., Xi’an 710065, China; 4Sichuan Chuanmei Huarong Energy Co., Ltd., Panzhihua 617000, China

**Keywords:** gob-side entry retaining, cyclical loading, back-filling concrete, damage evolution, energy dissipation

## Abstract

The back-filling body in the gob-side entry retaining is subject to continuous disturbance due to repeated mining. In this study, uniaxial and cyclical loading tests of back-filling concrete samples were carried out under laboratory conditions to study damage evolution characteristics with respect to microscopic hydration, deformation properties, and energy evolution. The results showed that, due to the difference in the gradation of coarse and fine aggregates, the cemented structure was relatively loose, and the primary failure modes under cyclical loading were tensile and shearing failure, which significantly decreased its strength. With an increasing number of loadings, a hysteresis loop appeared for the axial strain, and the area showed a pattern of decrease–stabilization–increase. This trend, to a certain extent, reflected the evolution of the cracks in the back-filling concrete samples. The axial, radial, and volumetric plastic strain curves of the back-filling concrete samples showed a “U” shape. The plastic strain changed in three stages, i.e., a rapid decrease, stabilization, and a rapid increase. A damage parameter was defined according to the plastic strain increment to accurately characterize the staged failure of the samples. The plastic strain and energy dissipation of the samples were precursors to sample failure. Prior to the failure of the back-filling samples, the amount and speed of change of both the plastic strain and energy parameters increased significantly. Understanding the characteristics of plastic strain, damage evolution, and energy dissipation rate of the back-filling samples are of great reference value for realizing real-time monitoring of back-filling concrete in the gob-side entry retaining and providing early warning of failure.

## 1. Introduction

Gob-side entry retaining technology is one of the key directions of development in pillar-free coal mining. Concrete is the most common roadside back-filling material due to its characteristics of fast curing, high load-bearing capacity, and simple process [1]. The back-filling body in the gob-side entry retaining is subject to repeated loads such as mining disturbance and roof breakage; thus, it undergoes a cyclical loading and unloading process. Due to damage accumulation and energy dissipation, stiffness is degraded and stress redistributes, reducing the load-bearing performance of the concrete material and bringing great difficulty to the stability control of the surrounding rock in the gob-side entry retaining [2,3,4,5].

Many studies have been carried out on the stress state and deformation mechanism of the back-filling body in the gob-side entry retaining under the influence of mining activities. Huang et al. [6] monitored the pressure of the roadside back-filling body in the gob-side entry retaining and analyzed the deformation and stress characteristics of the roadside back-filling body. Kan et al. [7] established the continuous laminate model under different roof conditions and proposed an equation to calculate the supporting resistance of the roadside back-filling body in the gob-side entry retaining. Feng et al. [8] reported that the stress and deformation of the roadside backfill body for gob-side entry of fully mechanized caving in thick coal seams were directly related to the breaking of the working face roof. The stress first increased and then stabilized with the advancement of the working face. Meng et al. [9] analyzed the roadside backfill body in gob-side entry retaining under combined static and dynamic loading and discussed the influence of the roof cutting angles on the behavior of the roadside backfill body using discrete element methods. Fan et al. [10] proposed an innovative BCR-GER approach for the complex geostress environment of deep coal seams based on the mechanical analysis of the roadside backfill body and the surrounding rock. The results from the above studies regarding the stress distribution and deformation of the back-filling body provide reference points for the design of roadway support in the gob-side entry retaining. However, implementing supporting protection in the deep gob-side entry retaining in recent years has shown that the back-filling body is often subject to repeated disturbance of roof subsidence in the current section and advancement of the working face in the lower section, causing nonlinear deformation, severe rock pressure, and failure of the back-filling body. Yet, few studies have focused on the damage or instability mechanism of the back-filling body in the gob-side entry retaining under cyclical loading conditions.

A large number of studies test different rock materials under cyclical loading conditions. For instance, Deng et al. [11] carried out a uniaxial cyclical loading test for sandstone. Considering the influence of residual deformation and the hysteresis effect, the authors modified the calculation method of energy parameters during the cyclical loading process. The results showed that rapid changes in energy parameters and residual strain were able to predict failure in the sandstone samples. Using digital image correlation technology, Yang et al. [12] defined a non-uniform deformation index and analyzed the evolution of non-uniform deformation and localized characteristic parameters during cyclical loading. Li et al. [13] carried out uniaxial cyclical loading testing of red sandstone and obtained the staged evolution characteristics of axial deformation, elastic energy, and dissipated energy. Wu et al. [14] analyzed the evolution of rock energy under different graded cyclical loading and unloading modes and established an evolution equation between dissipated energy, the cyclic stress level, and the number of loading cycles. This study laid a solid foundation for analyzing rock damage under cyclical loading based on energy dissipation. Yu et al. [15] analyzed the volumetric strain and volume expansion characteristics of marble fractures using variable amplitude cyclical loading tests under different confining pressures. These authors found that with increased loading times, the initial volumetric strain of marble fractures increased, and there was an exponential relationship between the increment of volumetric strain in a single loading cycle and stress amplitude. Tang et al. [16] studied local micro-cracks and nonlinear energy evolution of limestone under variable amplitude cyclical loading. The results indicated that both the ratio of elastic energy to total energy and the ratio of dissipated energy to total energy show staged characteristics. Variation in dissipated energy was related to the distribution of prefabricated cracks, the spatial location of micro-cracks, and the expansion rate. In addition, through triaxial cyclical loading tests of granite, Miao et al. [17] studied the evolution characteristics of dissipation energy, friction energy dissipation, and crushing energy dissipation under cyclical loading conditions. These authors found that the rock damage variable based on crushing energy dissipation could reasonably describe the damage evolution process of granite under cyclical loading.

In terms of the mechanical properties of concrete, Breccolotti et al. [18] proposed a constitutive model capable of accurately describing the damage accumulation of plain concrete subjected to cyclic uniaxial compressive loading. Park et al. [19] analyzed the microstructure and elastic–plastic characteristics of concrete subjected to cyclical loading using a nonlinear resonant ultrasonic method and quantitatively characterized the nonlinearity variation and load damage of concrete. Song et al. [20] studied the energy dissipation characteristics of concrete subjected to uniaxial cyclical loading based on dissipated energy and found that energy dissipation and damage evolution were stress-path-dependent. Hu et al. [21] analyzed the structural performance of recycled aggregate concrete subjected to cyclical loading, identified the characteristic points pertaining to the hysteresis loop, and proposed a simplified constitutive equation. Hutagi et al. [22] studied the stress–strain characteristics of geopolymer concrete (GPC) under cyclical loading; the analytical equations of the envelope curve, the common point curve, and the stable point were proposed. From the above studies, it is notable that there is little published work on the prediction of damage accumulation, progressive failure, and instability of back-filling concrete subjected to cyclical loading.

There are significant differences in rock deformation and energy evolution under cyclical loading compared to conventional uniaxial loading. Therefore, this article selects the back-filling concrete material as the research object, considering the repeated mining action, designs a continuous cyclic loading test, explores and characterizes the back-filling concrete material fracture extension, damage accumulation effect, and energy dissipation mechanism of back-filling concrete subjected to cyclical loading, and reveals the deep gob-side entry retaining deformation failure mechanism. It provides a basis for the monitoring and stability analysis of back-filling body in the gob-side entry retaining, which has important practical engineering value.

## 2. Materials and Methods

### 2.1. Sample Preparation

The back-filling concrete was composed of Portland cement, river sand, crushed stone, and a high-efficiency water-reducing agent. The cementing material was Jidong 42.5 ordinary Portland cement, and the coarse aggregate was crushed stone with a particle size of 5–16 mm and a density of 2.72 kg/m^3^. The fine aggregate was medium sand with a fineness modulus of 2.6–3.0 and a density of 2.6 g/cm^3^. The high-efficiency water-reducing agent was a polycarboxylic acid-type water-reducing agent, with a surface density of 550–650 G/L, an active ingredient of greater than 99%, and a pH value (10% aqueous solution) between 7.0 and 8.0.

According to the formula for concrete in the gob-side entry retaining (cement: river sand (0–5 mm): crushed stone (5–15 mm): water = 1:0.82:1:0.38), the materials were mixed and stirred well before being poured into a mold. The mixture was vibrated sufficiently to reduce air bubbles, and then the surface was covered with fresh film to prevent the surface from drying and cracking, curing it with a mold. The concrete sample mold was placed in the standard curing box and the temperature was set to 20 ± 2 °C; while the relative humidity was 90% after curing for 28 days. According to the recommendations of the International Association of Rock Mechanics, the concrete samples were formed into a cylindrical shape with a diameter of 50 mm and a height of 100 mm, and the diameter error was controlled to within 0.3 mm (Samples No. J1–J20). Moreover, both ends of the samples were ground such that surface flatness was within 0.02 mm. In addition, an ultrasonic test device was used to test the wave velocity of the samples, and those with comparable wave velocities were chosen for testing. At a normal temperature, the average longitudinal wave velocity of the samples was 2147 m/s, and the average density was 2.17 g/cm^3^.

### 2.2. Test Scheme

Based on the mechanical conditions of the back-filling body in the gob-side entry retaining, a cyclical loading test was set up in an indoor laboratory. The linear correlation between the compressive strength and the ultrasonic wave velocity was established based on the results of uniaxial loading (Figure 1). The loading path is shown in Figure 2. First, the compressive strength *σ_c_* of the samples was first predicted based on the linear correlation ($\sigma_{c}=0.9358v-1982.6$) and the initial wave velocity, then the upper and lower limit of the cyclical load was determined (*σ*_min_ = 0.4*σ*_c_, *σ*_max_ = 0.8*σ_c_*), as shown in Table 1. The load was applied under force control and the loading process was as follows: (1) In the first stage, the sample was linearly loaded at a speed of 0.2 kN/s to σ_0_ = 0.6*σ_c_*; (2) in the second stage, sine wave cyclical loads were applied to the sample (amplitude: 0.4–0.8*σ_c_*, frequency: 0.4 Hz) until sample failure. Ultrasonic testing was carried out continuously throughout the entire loading cycle. After data processing, the parameters related to the strength, deformation, and failure of the sample and the energy evolution under cyclical loading were obtained.

### 2.3. Equipment

The MTS815 material testing machine was used via electro-hydraulic driven operation (Figure 3). The axial load capacity was 2300 kN, the upper limit of confining pressure was 140 MPa, and the accuracy of the pressure transducer was 0.001 kN. Deformation was measured with an axial extensometer and a circumferential extensometer. The measuring range of the extensometers was between 5 mm and 8 mm, respectively, and the resolutions were both 10^−4^ mm. Through the Flex Test GT digital controller, the MTS machine was able to realize load and displacement control modes.

An ultrasonic system was used for damage accumulation monitoring (Figure 3). The transmitting and receiving transducers were arranged on the upper and lower ends of the samples, and a coupler was applied to ensure good contact between the sample and the transducer. The resonance frequency (HYN, 55 KHz) and damping meet the requirements for mechanical testing of concrete materials. A waveform generator actively transmitted a pulse signal (main frequency: 90 kHz; amplitude: 10 Vpp). After receiving the pulse signal, one end of the transducer generated instantaneous vibration. The vibration propagated in the sample and was then received by the transducer on the other end. The data acquisition frequency was 10 Mpoints/s, and the duration was 5 ms.

The Quanta450& IE250X-MAX50 scanning electron microscope was used to scan the microstructure of back-filling concrete samples. Block samples were taken from back-filling concrete samples of different ages and fixed with conductive adhesive. In order to enhance the electrical conductivity of back-filling concrete, gold was sprayed on its surface.

## 3. Results and Discussion

### 3.1. Deterioration of Load-Bearing Performance of Concrete Samples Subjected to Cyclical Loading

(1)Sample microstructures

The microscopic morphology of the back-filling concrete samples with different curing periods is shown in Figure 4 (magnification 5000×). At 7 days of curing, a small amount of hydration product calcium-silicate-hydrate (C-S-H) particles was seen around the aggregate, as shown in Figure 4a. Note that the longer the curing period of the back-filling concrete, the better the crystallinity of the hydration product. Figure 4b shows that the surface was rough, containing particles bound by C-S-H, and the structure of the concrete samples was relatively loose at 14 days. Increasing the curing period gradually increases the amount of hydration product, and a large number of links formed between C-S-H and the aggregates, while the internal pores were continuously filled, which enhanced the strength of the aggregate–cement interface, as shown in Figure 4c,d. Overall, a large number of micropores and cracks were formed in the concrete samples, which had strong heterogeneity.

(2)Macroscopic fracture characteristics

The fracture patterns of the samples under different loading paths are shown in Figure 5a,b. In Figure 5a, the red lines represent the main cracks and the thin black line represents the secondary cracks. When the loading path was changed from uniaxial loading to constant-amplitude cyclical loading, the main cracks were connected upon sample failure, accompanied by a large number of secondary cracks. The number and density of macroscopic cracks increased significantly in Figure 5b. Specifically, a large number of cracks appeared parallel to the principal stress direction. The shear failure zone showed a large number of fragments and localized exfoliation. 

During the loading process, lateral deformation of the sample was large, and there was clear volumetric expansion. This was mainly because the load did not decrease to 0 kN during the unloading phase, and damage accumulated under cyclical loading. When the samples were damaged, a macroscopic crack penetrating the whole sample appeared, and the failure modes were mainly tensile and shearing failure. In addition, the peak ultimate failure load of the samples decreased significantly under cyclical loading. The intensity reduction ratio was 20.99–31.49% of that obtained during uniaxial compressive loading (Table 1), i.e., the load-bearing performance of the concrete sample declined significantly.

### 3.2. Deformation Characteristics of Concrete Samples Subjected to Cyclical Loading

(1)Stress–strain curve

As Table 1 shows, the total number of loading cycles of the J-4 sample was 55 times and the J-19 sample was 95 times in the two groups’ samples, and J-4 and J-19 are selected as the representatives of both groups of samples. The stress–strain curve of the concrete samples subjected to cyclical loading is shown in Figure 6a,b, The failure stress of the concrete samples was relatively close, and the stress–strain curves showed hysteresis loops. With the increase in the loading cycle, the area of the hysteresis loop gradually increased. The reason for this was that each loading cycle led to new damage to the internal samples, and the damage continued to accumulate, resulting in a continuous increase in the axial strain on the sample. The lower load limit was above 0 kN, thus, the micro-cracks generated during cyclical loading were always in a compressive state, and the cumulative plastic strain gradually increased. Compared with J19 in Figure 6b, sample J4 exhibited more obvious ductility during fatigue failure as shown in Figure 6a.

With the increased loading cycle, irreversible deformation of the concrete samples increased and showed a nonlinear increasing trend. Macroscopically, the sample showed clear volumetric expansion and fatigue failure.

(2)Plastic hysteresis loop

Based on the maximum hysteresis loop area before failure, the area of the plastic hysteresis loop in each loading cycle was normalized. Taking the ratio of the total number of loading cycles to the number of cycles at fatigue failure as the *x*-axis, *n_i_/n* represents the total number of loading cycles to the number of cycles at fatigue failure, and the variation of the area of the plastic hysteresis loop with the number of loading cycles is obtained (Figure 7). Figure 7a notes that as the number of loading cycles increased, the area of the plastic hysteresis loop at each time showed a staged pattern of decrease–stabilization–rapid increase and showed a “U” shape. The loading stress in the first loading cycle (*σ*_max_ = 0.8*σ_c_*) exceeded the yield strength of the concrete material, and the sample entered the plastic deformation stage. There was a large amount of plastic deformation, indicated by the large area of the plastic hysteresis loop. After unloading, the elastic deformation recovered, and the area of the plastic hysteresis loop decreased significantly.

With the progression of cyclical loading, the cracks in the samples were in a constant compressive state, and the plastic strain continuously accumulated, although the sample was still in a stable condition. In the later stage of cyclical loading, the area of the plastic hysteresis loop in a single cycle gradually increased with increasing amplitude, indicating an increasing degree of fatigue damage. Once macroscopic cracks penetrated the sample, the load-bearing performance greatly declined.

Figure 7b shows that the area of each hysteresis curve was normalized, and the normalized cumulative hysteresis loop area showed a nonlinear increase with the increase in the number of cycles, and its growth rate showed a trend of “decrease–stable–increase”.

(3)Variation of plastic strain

The elastic strain $\varepsilon_{e}$, plastic strain $\varepsilon_{p}$, and volumetric strain $\varepsilon_{v}$ of the concrete samples subjected to cyclical loading were calculated to quantify the evolution of the plastic strain. The physical implications of elastic and plastic strains are shown in Figure 8, and the elastic and plastic strains are calculated as [23]:(1)$$\varepsilon_{e}=\varepsilon_{\max}-\varepsilon_{\min}^{}$$
(2)$$\varepsilon_{p}=\varepsilon_{\min}^{}-\varepsilon_{\min}^{\prime }=\varepsilon_{\min}-\frac{\sigma_{\min}}{E}$$
where $\varepsilon_{e}$ and $\varepsilon_{p}$ are the elastic strain and plastic strain, respectively; $\varepsilon_{\max}$ is the strain corresponding to the upper load limit, *E* is the elasticity modulus, $\varepsilon_{\min}$ represents the strain corresponding to the lower load limit, and $\varepsilon_{\min}^{\prime }$ represents the elastic strain corresponding to the lower load limit.

Under uniaxial cyclical loading, assuming the axial strain is $\varepsilon_{1}$ and the radial strain is $\varepsilon_{3}$, the volumetric strain is calculated as [24]:(3)$$\varepsilon_{v}=\varepsilon_{1}+2\varepsilon_{3}$$
where $\varepsilon_{v}$ is the volumetric strain, $\varepsilon_{1}$ is the axial strain, and $\varepsilon_{3}$ is the radial strain.

Figure 9a–f show the relationship between the axial strain, radial strain, and volumetric plastic strain of some samples (J-4, J-11, J-13, J-14, J-17, J-18, J-19, and J-20) and the number of loading cycles.

Note from Figure 9 that the variation of the axial, radial, and volumetric strains with the number of loading cycles was essentially consistent, showing a “U” shape. The plastic strain presented a staged pattern of a rapid decrease–stabilization–a rapid increase. Thus, the variation of plastic strain showed clear stage characteristics. *n* is the total cycle number of each sample, the cycle number of the demarcation point between the rapid decrease and stabilization stages was set as *n*_1_, and the cycle number of the demarcation point between the stabilization stage and the rapid increase stage was set as *n*_2_. The *n*_1_ and *n*_2_ values of each sample are shown in Table 2.

As one can see from Figure 9a,c, in the early stage, the loading stress was above the yield stress, which led to a large amount of plastic deformation. After unloading, the elastic deformation was recovered, and the axial and radial deformations decreased significantly. In the late stage, the internal cracks propagated due to repeated compression and the degree of fatigue damage increased, as indicated by the increasing axial and radial plastic deformation. Then, cracks penetrated the sample, and the plastic deformation increased dramatically to the maximum value, and the sample underwent shearing-tensile failure, and significant radial cracks developed.

With increasing loading cycles, the plastic strain gradually accumulated, yet the sample was still in a stable state. The accumulative velocity of axial plastic strain first increases and then decreases with the increase in the loading cycles as shown in Figure 9b,d. During the cyclical loading process, the volumetric strain showed a significant nonlinear increasing trend as shown in Figure 9e. In the initial stage, the volumetric plastic strain of the J-4, J-13, and J-14 samples had a positive increment, indicating that axial compressive deformation was much larger than radial expansion deformation. However, the positive increment was small and rapidly became negative, indicating that there were few initial cracks in the concrete sample and micro-cracks were initiated. Furthermore, the volumetric strain gradually changed from axial compression deformation to radial expansion deformation. For the J-11, J-17, J-18, J-19, and J-20 samples, the initial volumetric strain had negative increments, showing clear radial volumetric expansion as shown in Figure 9f. As cyclical loading continued, the internal micro-cracks were repeatedly squeezed and propagated, and the proportions of axial, radial, and volumetric plastic strains all gradually increased. In addition, the growth rate of the plastic strain increased sharply before sample failure (80.00–85.19% of the total number of cycles) as shown in Table 2. As a result, primary and secondary cracks were connected, and the concrete sample was damaged.

### 3.3. Energy Dissipation of Concrete Samples Subjected to Cyclical Loading

(1)Strain energy density

During the cyclical loading process, the samples exchanged energy with the environment. As the loading increased on the concrete samples, the accumulated strain energy was continuously transformed and dissipated, accompanied by the deformation and failure of the samples. In the stage before peak stress, energy conversion involved only elastic energy storage and energy dissipation. The total strain energy density, *W*, is the sum of the elastic strain energy density, *W**_e_*, and the energy dissipation density, *W**_d_*, which can be expressed as [25,26]:(4)$$W=W_{e}+W_{d}$$
(5)$$W=\int_{0}^{\varepsilon_{i}}\sigma_{i}d\varepsilon_{i}$$
(6)$$W_{e}=\int_{\varepsilon_{p}}^{\varepsilon_{i}}\sigma_{i}^{\prime }d\varepsilon_{i}^{\prime }$$
where $\sigma_{i}$ and $\sigma_{i}^{\prime }$ are the loading and unloading stress of the *i*-th cycle, respectively; $\varepsilon_{i}$ and $\varepsilon_{i}^{\prime }$ are the axial strain in the loaded and unloaded state in the *i*-th cycle, respectively.

(2)Energy evolution characteristics

Figure 10 shows that elastic strain energy first increased with the increasing loading cycles, then stabilized with small fluctuations, then decreased rapidly before sample failure. Furthermore, the total strain energy and dissipated energy of the concrete samples had similar trends with obvious stage characteristics under cyclical loading. The total strain energy and dissipated energy were high in the early stage due to the first cycle having higher loading and deformation; with increasing loading cycles, they decreased first and then stabilized and increased rapidly in the late stage. The slope change of curve was taken as the basis for stage. The turning point of dissipated energy rapidly decreases to stability with small fluctuations was taken as the stage segmentation point from stage I to II, such as Figure 10a (6,0.80) and 10b (5,0.36). The turning point of dissipated energy steadily develops to a rapid increase was taken as the stage segmentation point from stage II to III, such as Figure 10a (41,0.75) and 10b (81,0.38). And Failure of the concrete samples was accompanied by a large increase in dissipated energy.

To accurately describe the damage evolution process of the concrete samples subjected to cyclical loading and to characterize the energy dissipation, the variation in the energy dissipation rate with the number of cycles for different samples is shown in Figure 11. The *x*-axis is the total number of loading cycles in Figure 11a, the *x*-axis is the ratio of the total number of loading cycles to the number of cycles at fatigue failure in Figure 11b, and the *y*-axis is the energy dissipation rate. The ratio of energy dissipation density (*W**_d_*) to total strain energy (*W*) was defined as the energy dissipation rate [26].
(7)$$W_{k}=\frac{W_{d}}{W}$$
where *W_k_* is the energy dissipation rate.

Note from Figure 11 that the energy dissipation rate shows a pattern of sharp decrease–steady fluctuation–rapid increase with increased loading cycles. Take J-4 sample for example, the slope change of energy dissipation rate curve was taken as the basis for three stages. The turning point of energy dissipation rate sharp decrease to steady fluctuation was taken as the point from stage I to II, such as Figure 11a (6,0.07). The turning point of energy dissipation rate steadily develops to a rapid increase was taken as the point from stage II to III, such as Figure 11a (42,0.07).

According to the slope of the energy dissipation rate curve, the energy dissipation process can be divided into three stages (I, II and III):

Initial dissipation stage I (0–10% of the cycle): In the early stage of cyclical loading, the loading stress in the first cycle exceeded the yield stress, resulting in large plastic deformation and crack development. Thus, energy dissipation was large, and the plastic strain energy rate accounted for 30% of the total strain energy. After the first cycle, the number of new micro-cracks gradually decreased, the plastic deformation increment decreased sharply, and the plastic strain energy decreased gradually. The energy dissipation rate decreased sharply, while the accumulated elastic strain energy increased gradually.

Stable dissipation stage II (10–80% of the cycle): As cyclical loading continued, the lower stress limit in the unloaded state did not return to zero, there was compaction of internal pores and cracks and propagation of micro-cracks, and plastic deformation continued increasing with a small increment. The energy dissipation rate was maintained at 0.02–0.08 at this stage, and there was a small fluctuation. Most of the energy was converted into elastic strain energy, and energy dissipation was small.

Rapid dissipation stage III (80–100% of the cycle): With increased loading cycles, the micro-cracks gradually propagated to form large cracks, and the large cracks gradually penetrated to form macro-cracks. Plastic deformation increased rapidly at this stage, plastic strain energy density increased, and the energy dissipation rate increased significantly until the sample failed.

As shown in Figure 11a, before the failure of the J-4 sample, the amount of change and the change in speed of both the plastic strain and energy parameters increased significantly, indicating that the plastic strain and energy dissipation rates could predict the failure of the concrete samples.

## 4. Conclusions

(1)Back-filling concrete samples were prepared using Portland cement, river sand, crushed stone, and a high-efficiency water-reducing agent. With an increased curing period, hydration products increased, pores and fractures decreased, and the strength of the aggregate cement interface increased continuously, which has the characteristics of the fast lifting of the bearing strength and strong ductility. Due to the heterogeneity of the mesostructure, the failure modes were mainly tensile and shear failure, and the load-bearing performance decreased significantly after failure.(2)Under cyclical loading, the stress and strain curves of the concrete samples show obviously staged characteristics. With increased loading cycles, the evolution of plastic hysteresis loops shows the characteristic of “sparse–dense–sparse”, the area of the hysteresis loops showed a pattern of decrease–stabilization–rapid increase, and the larger the hysteresis loop area, the more obvious the volume expansion and the more obvious the bearing capacity reduction. Furthermore, the axial, radial, and volumetric plastic strain showed a “U” shape, and the plastic strain presented a staged pattern of a rapid decrease–stabilization–rapid increase, which directly reflects the regularity of fracture development and damage accumulation evolution in back-filling concrete samples under the influence of repeated mining.(3)Variation in the dissipated energy of the concrete sample subjected to cyclical loading was closely related to micro-crack propagation and damage evolution. The elastic energy increases with the increase in cyclic loading and unloading times, and the energy dissipation process could be divided into three stages: The initial dissipation stage, the stable dissipation stage, and the rapid dissipation stage. Furthermore, the dissipated energy first decreased, then stabilized with small fluctuations, and then increased rapidly. The calculation method of the dissipative energy rate is introduced to accurately describe the phase-change law of dissipative energy accumulation, which provides a basis for predicting the failure precursor of back-filling concrete samples.

## Figures and Tables

**Figure 1 materials-15-05772-f001:**
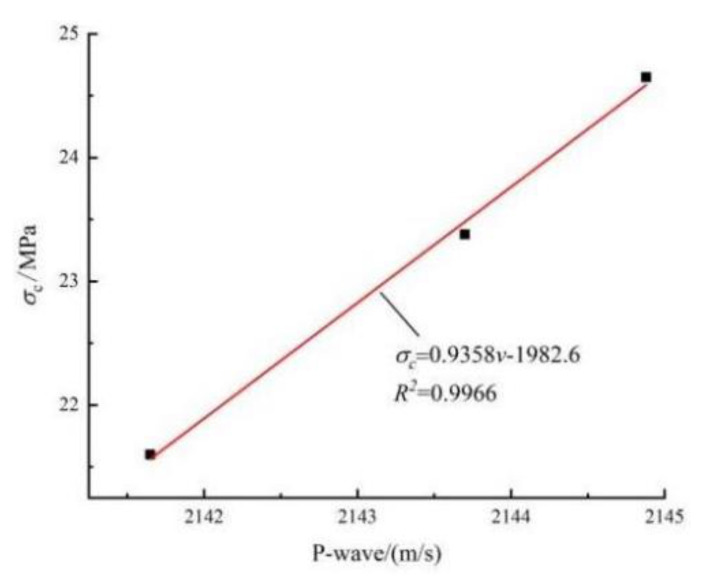
Relationship between uni-axial compressive strength and p-wave velocity.

**Figure 2 materials-15-05772-f002:**
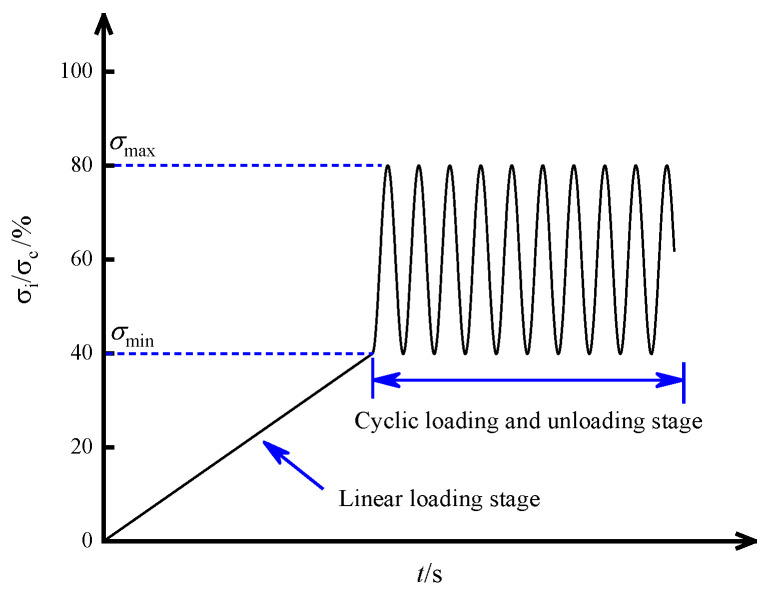
Cyclic loading method.

**Figure 3 materials-15-05772-f003:**
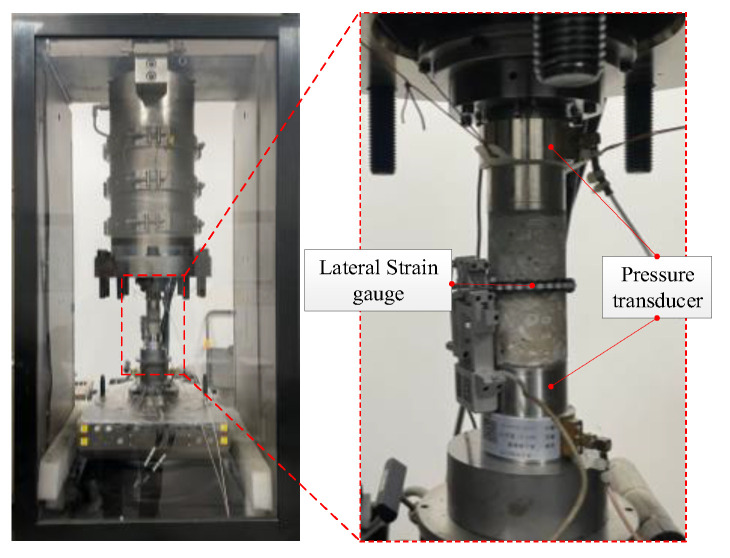
Sample loading device and ultrasonic pressure transducer layout.

**Figure 4 materials-15-05772-f004:**
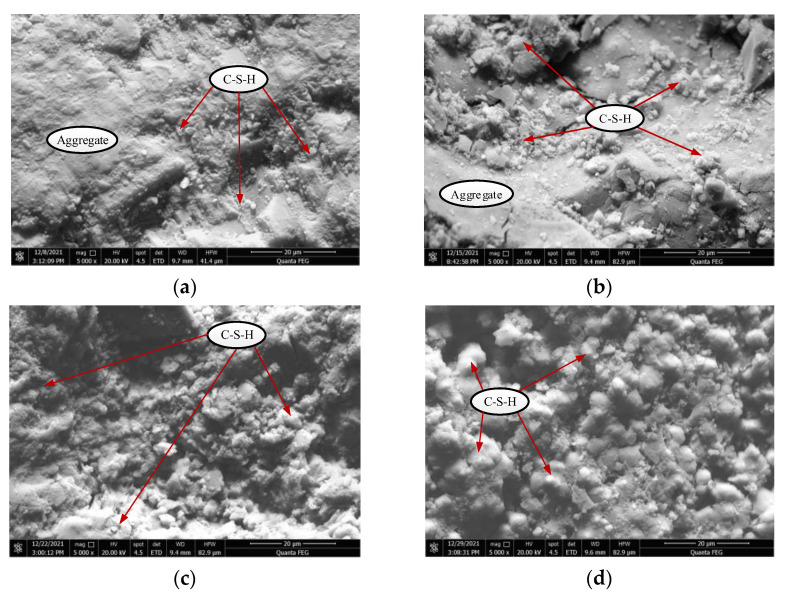
Scanning electron microscope of back-filling concrete at different ages. (**a**) 7 days; (**b**) 14 days; (**c**) 21 days; (**d**) 28 days.

**Figure 5 materials-15-05772-f005:**
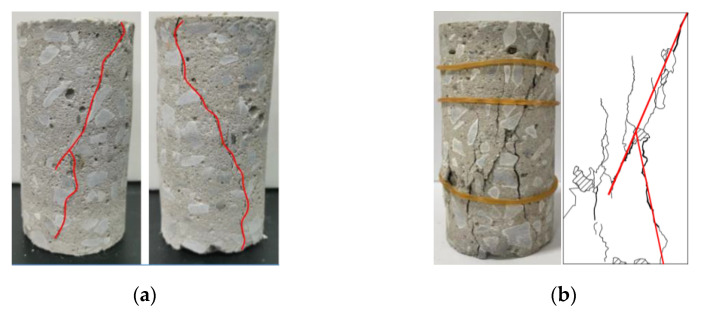
Fracture characteristics of back-filling concrete samples under different loading paths. (**a**) J-1 sample uniaxial loading path; (**b**) J-11 sample under cyclic loading path.

**Figure 6 materials-15-05772-f006:**
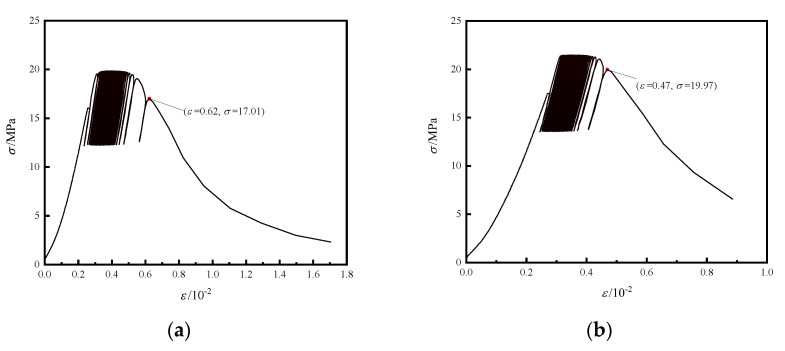
Stress–strain curves of cyclic loading and unloading concrete samples. (**a**) J-4; (**b**) J-19.

**Figure 7 materials-15-05772-f007:**
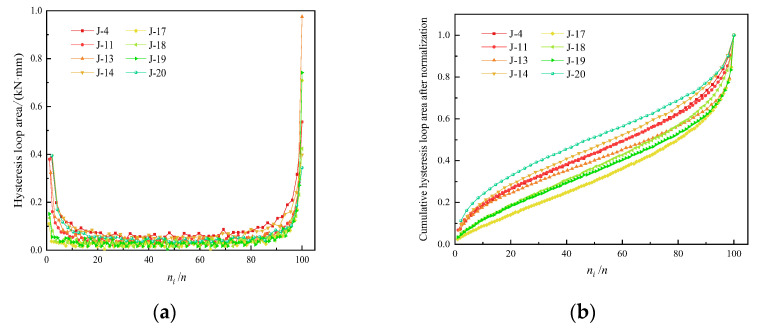
Variation rule of hysteresis loop area of cyclic loading and unloading. (**a**) Hysteresis loop area each cycle. (**b**) Normalized cumulative hysteresis loop area.

**Figure 8 materials-15-05772-f008:**
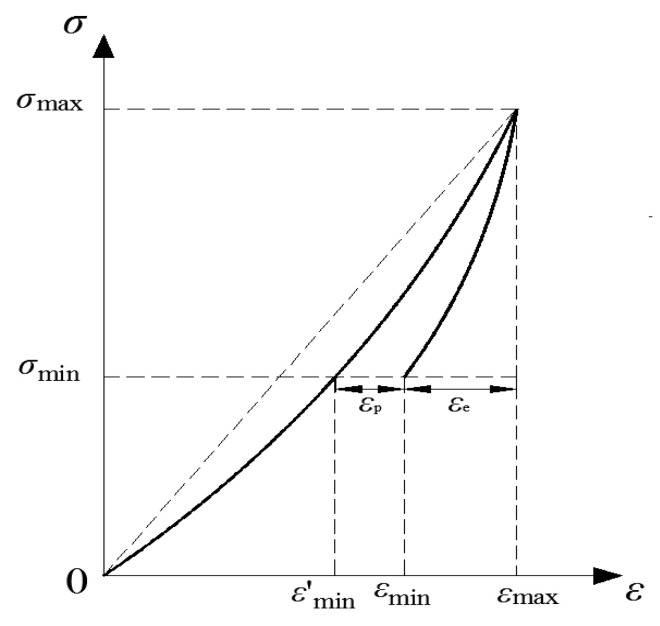
Schematic diagram of strain parameters.

**Figure 9 materials-15-05772-f009:**
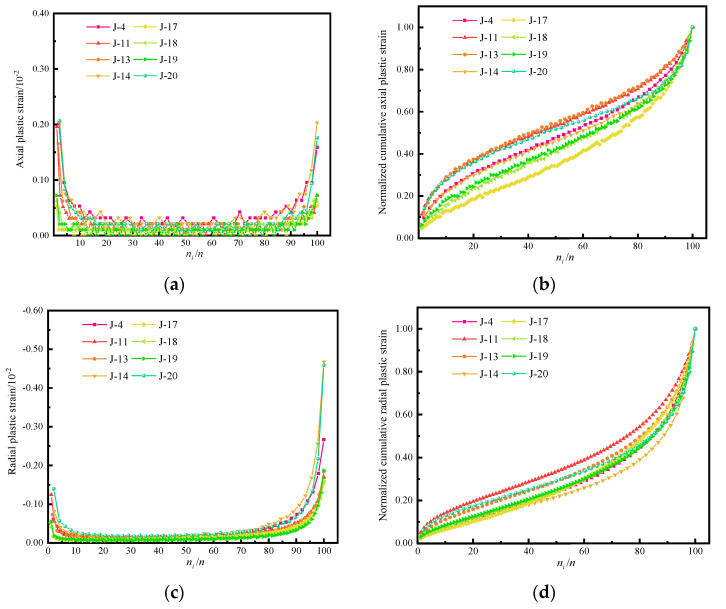
Variation curve of plastic strain with cyclic loading and unloading times. (**a**) Axial plastic strain; (**b**) normalized accumulative axial plastic strain; (**c**) radial plastic strain; (**d**) normalized accumulative radial plastic strain; (**e**) volumetric plastic strain; (**f**) normalized accumulative volumetric plastic strain.

**Figure 10 materials-15-05772-f010:**
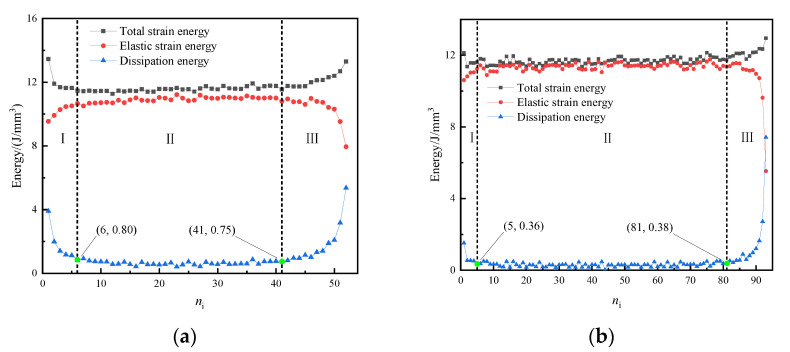
Energy evolution curve of back-filling concrete sample. (**a**) J-4; (**b**) J-19.

**Figure 11 materials-15-05772-f011:**
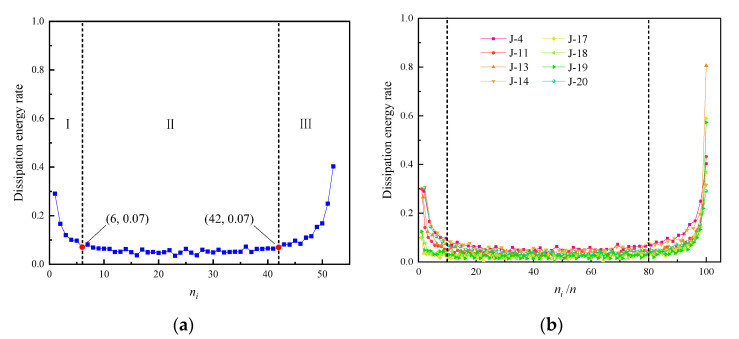
Variation of dissipation energy rate with the number of cycles. (**a**) Dissipation energy rate change curve of J-4 sample. (**b**) Stage division of dissipation energy rate of samples.

**Table 1 materials-15-05772-t001:** Cyclic loading and unloading test parameters.

Sample	Prediction of Uniaxial Compressive Strength (MPa)	Cycle Loading Times	Failure Stress (MPa)	Intensity Decay Ratio (%)
J-4	24.83	55	17.01	31.49
J-13	23.83	64	17.00	28.66
J-14	24.83	48	17.57	29.24
J-20	23.82	50	18.82	20.99
J-11	24.82	90	18.81	24.21
J-17	24.83	108	18.52	25.41
J-18	26.82	90	19.98	25.50
J-19	25.69	95	19.97	22.27

**Table 2 materials-15-05772-t002:** Node classification of plastic strain stage.

Samples	*n*	*n* _1_	*n*_1_/*n*	*n* _2_	*n*_2_/*n*
J-4	55	6	10.91%	45	81.82%
J-11	90	11	12.22%	76	84.44%
J-13	64	9	14.06%	53	82.81%
J-14	48	6	12.50%	39	81.25%
J-17	108	11	10.19%	92	85.19%
J-18	90	10	11.11%	72	80.00%
J-19	95	10	10.53%	80	84.21%
J-20	50	6	12.00%	41	82.00%
average	/	/	11.69%	/	82.72%

## Data Availability

The data used to support the findings of this study are included in the article.

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
