# Peer review of "Damage Evolution Characteristics of Back-Filling Concrete in Gob-Side Entry Retaining Subjected to Cyclical Loading"

_materials, 2022, doi:10.3390/ma15165772_

Round 1

Reviewer 1 Report

Dear authors,

My comments are attached in the file below-Review Materials_1840313.

Sincerely

Reviewer 3 Report

Title of paper: “Damage Evolution Characteristics of Back-Filling Concrete in Gob-Side Entry Retaining Subjected to Cyclical Loading”

The research paper is interesting, but it needs to be improved. Here are some suggestions.

·         In Line no. 40 of the Introduction, the author used the word "energy dissipation," so what type of energy dissipation does the author focus on?

·         The author used the word "goaf" in Line 58 of the Introduction, which is incorrect because it is used in context. What did the author mean when he used that word?

·         Include a paragraph which summarises the various chapters that the author has defined. Include it in the last paragraph of the introduction.

·         2.1.1. Equipment Section The statement “MTS815 electro-hydraulic triaxial material testing machine was used” according to which code does this equipment as specification and adds the code reference in the references section.

·         Verify the manuscript in terms of prefixes and suffixes.

·         At Line nos. 153 and 154, the statement “The average initial wave velocity of the specimens included in this study was 2147 m/s. The density was 2170 kg/m3.” How did the authors arrive at this velocity and density value? What procedure did they follow?

·          Control Mix specimens is missing.

·         In line 166, what was the reason behind fixing the and amplitude: 0.4–0.8c, frequency: 0.4 Hz) in this experiment?

·         Write a paragraph about how you have defined materials, assigned the loads, and performed the analysis using software. In general, how did you perform analysis using software and include the details of the software?

·         Add a methodology flow chart for your work, List abbreviation and notations.

·         The test results are only for 28 days, but what about 7, 14 and 90-day results?

·         What is the significance of the different colours in figure 7?

·         Paragraph related to comparison between figure 9 (a to f), should be made.

·         4th and 6th equations Is this from the authors' analysis? If yes, then how do the authors arrive at the equation?

·         Conclusion must be specific rather than discussion.   

·         On Figure 4, authors have highlighted C-S-H. On what basis did authors come to know that portion belongs to C-S-H gel?

·          Specimen size is not mentioned in the manuscript.

·         What about the curing condition of specimens?

·         Sections 3.1, 3.2, and 3.3 support the discussion with a literature review. Add more references to the support reference section.: doi.org/10.1016/j.engfailanal.2021.105531; doi.org/10.3390/polym14020306 ; doi.org/10.1002/ese3.431

·         Overall, check the language of the paper and verify the results.

Round 2

Reviewer 1 Report

Dear authors,

You fixed everything I asked for, however, I have a few more technical objections.

1.       Line 122….2.1.sample preparation should be fixed in 2.1. Sample preparation

2.       The numbers and letters in Fig. 1 are unclear- should be fixed

3.       Line 188…..(1) sample microstructure should be fixed in (1) Sample microstructure

4.       Line 191……should be (Calcium-Silicate-Hydrate)

5.       Line 205…Fig. 4 Scanning electron microscope of back-filling concrete at different age: (a) 7 days; (b) 14 days; (c) 21 days; (d) 28 days

Under the figures, write only (a), (b), (c), (d)

6.       Fig. 8 is small, it needs to be enlarged. 

7.       Please, carefully review the list of references again.

Sincerely

Reviewer 3 Report

The author has been revised according to the review comments to meet the publication requirements.